# The Dual Lens of Endoscopy and Histology in the Diagnosis and Management of Eosinophilic Gastrointestinal Disorders—A Comprehensive Review

**DOI:** 10.3390/diagnostics14080858

**Published:** 2024-04-22

**Authors:** Alberto Barchi, Edoardo Vespa, Sandro Passaretti, Giuseppe Dell’Anna, Ernesto Fasulo, Mona-Rita Yacoub, Luca Albarello, Emanuele Sinagra, Luca Massimino, Federica Ungaro, Silvio Danese, Francesco Vito Mandarino

**Affiliations:** 1Gastroenterology and Digestive Endoscopy, IRCCS Ospedale San Raffaele, Via Olgettina 58, 20132 Milan, Italy; barchi.alberto@hsr.it (A.B.); vespa.edoardo@hsr.it (E.V.); passaretti.sandro@hsr.it (S.P.); dellanna.giuseppe@hsr.it (G.D.); fasulo.ernesto@hsr.it (E.F.); massimino.luca@hsr.it (L.M.); ungaro.federica@hsr.it (F.U.); danese.silvio@hsr.it (S.D.); 2Unit of Immunology, Rheumatology, Allergy and Rare Diseases, IRCCS Ospedale San Raffaele, 20132 Milan, Italy; yacoub.monarita@hsr.it; 3Pathology Unit, IRCCS Ospedale San Raffaele, 20132 Milan, Italy; albarello.luca@hsr.it; 4Gastroenterology and Endoscopy Unit, Fondazione Istituto S. Raffaele—G. Giglio, 90015 Cefalu, Italy; emanuelesinagra83@googlemail.com; 5Faculty of Medicine, Università Vita-Salute San Raffaele, 20132 Milan, Italy

**Keywords:** EGIDs, eosinophilic GI disorders, eosinophilic esophagitis, eosinophilic colitis

## Abstract

Eosinophilic Gastrointestinal Disorders (EGIDs) are a group of conditions characterized by abnormal eosinophil accumulation in the gastrointestinal tract. Among these EGIDs, Eosinophilic Esophagitis (EoE) is the most well documented, while less is known about Eosinophilic Gastritis (EoG), Eosinophilic Enteritis (EoN), and Eosinophilic Colitis (EoC). The role of endoscopy in EGIDs is pivotal, with applications in diagnosis, disease monitoring, and therapeutic intervention. In EoE, the endoscopic reference score (EREFS) has been shown to be accurate in raising diagnostic suspicion and effective in monitoring therapeutic responses. Additionally, endoscopic dilation is the first-line treatment for esophageal strictures. For EoG and EoN, while the literature is more limited, common endoscopic findings include erythema, nodules, and ulcerations. Histology remains the gold standard for diagnosing EGIDs, as it quantifies eosinophilic infiltration. In recent years, there have been significant advancements in the histological understanding of EoE, leading to the development of diagnostic scores and the identification of specific microscopic features associated with the disease. However, for EoG, EoN, and EoC, precise eosinophil count thresholds for diagnosis have not yet been established. This review aims to elucidate the role of endoscopy and histology in the diagnosis and management of the three main EGIDs and to analyze their strengths and limitations, their interconnection, and future research directions.

## 1. Introduction

In the intricate landscape of gastrointestinal (GI) disorders, Eosinophilic Gastrointestinal Disorders (EGIDs) have emerged as a unique and intriguing group of conditions in recent years [1]. These conditions are characterized by abnormal eosinophil infiltration into the gastrointestinal wall [2]. The classification of EGIDs has evolved over time. Recently, an international consensus categorized these disorders based on the specific segment of the GI tract affected by the eosinophilic infiltration, using TIER 1 nomenclature for clinical use (Table 1) and TIER 2 nomenclature for research and clinical use (Table 2) [3].

Among EGIDs, Eosinophilic Esophagitis (EoE) is the most well documented, exclusive involving the esophagus, while Eosinophilic Gastritis (EoG) specifically affects the stomach and Eosinophilic Enteritis (EoN) is confined to the small intestine. Eosinophilic Colitis (EoC), the least understood and rarest of the group, is characterized by eosinophilic infiltration in the colon [4]. Each of these conditions exhibits unique clinical features, necessitating a tailored approach to diagnosis and treatment.

Eosinophils, which are integral components of the immune system, are normally present in various segment of the GI tract [5]. However, in the context of EGIDs, excessive eosinophilic infiltration leads to cell activation, resulting in active and potentially chronic inflammation. This causes persistent tissue damage and symptoms that vary according to the affected organ [6]. The pathophysiology behind this abnormal eosinophil accumulation is complex, involving a combination of genetic, environmental, and immune factors.

The management of EGIDs presents a significant challenge, given the complexity and often subtle nature of these conditions. Endoscopy plays an indispensable role in raising disease suspicion, guiding biopsy sampling, and monitoring disease progression and response to treatment [7]. Histological evaluation is the cornerstone of diagnosis, particularly for the quantification of eosinophils in tissue sections [8]. 

In EGIDs, the concentration of eosinophils in the tissue exceeds normal levels; however, these thresholds are not completely standardized, except for EoE, (Table 3). In all EGIDs, establishing a correlation between histological data, patients’ anamnesis, clinical history, and symptomatology is fundamental for accurate diagnosis.

This narrative review aims to comprehensively focus on the role of endoscopy and histology in the diagnosis and management of the three main EGIDs, analyzing their interconnections, strengths, relation to clinical and therapeutic aspects, and future scenarios.

### Pathophysiology of Eosinophilic Inflammation

Eosinophils, a specific type of leukocyte derived from CD34+ CD125+ stem cells in the bone marrow, are crucial in defending against pathogens, such as bacteria and parasites. Additionally, they play a pivotal role in modulating humoral immune IgA and cellular T-cell responses, and in maintaining tissue homeostasis [15]. 

Typically, eosinophils are dispersed throughout the GI tract, residing in the lamina propria, with the notable exception of the esophageal squamous epithelium [16,17,18,19]. Matsushita and colleagues noted that in healthy individuals, the concentration of eosinophils varies across different sections of the GI tract, generally increasing from the stomach to the distal small intestine [20]. In contrast, eosinophil distribution in the colon shows a descending gradient from the cecum to the distal colon [21]. 

In the context of EGIDs, the abnormal accumulation of eosinophils is primarily driven by interleukin-5 (IL-5), interleukin-4 (IL-4), and interleukin-13 (IL-13) (Figure 1). These cytokines are primarily produced by type 2 helper lymphocytes (Th2) in response to exposure to aeroallergens and food allergens. The overproduction of interleukins is further amplified by dysregulated cells in the innate immune system, including Group 2 innate lymphoid cells (ILC2s) that mature directly in tissues like the GI tract or lungs [22], plasma cells, and mast cells [18,19,23,24,25]. 

Th2 cytokines, particularly IL-13, along with other inflammatory mediators such as Tumor Necrosis Factor Alpha (TNF-α) and other chemokines, play a direct role in the activation and degranulation of eosinophils [26]. Eotaxin-3 serves as the primary chemokine involved in these processes and contributes significantly to eosinophilic chemotaxis and accumulation in GI tissues. The release of proteins from eosinophilic granules, including eosinophil cationic protein (ECP), eosinophil-derived neurotoxin (EDN), and major basic protein (MBP), leads to acute cytotoxic and oxidative damage to the tissue [14]. This acute damage results in compromised barrier function through the downregulation of Desmoglein 1 (DSG1), Filaggrin (FGN), and the Epidermal Differentiation Complex (EDC) [27].

The course of acute eosinophilic Th2 inflammation is typically self-sustained and progressive, often leading to chronic damage. This chronic state is often driven by T regulatory lymphocytes’ activation, accompanied by the recruitment of other cell types, including mast cells and basophils [26]. The persistent inflammatory insult can result in sub-mucosal fibrotic tissue deposition and muscular hypertrophy, primarily induced by Transforming Growth Factor beta (TGF-beta). In the esophagus, this process leads to anatomical remodeling characterized by reduced distensibility and compliance of the organ wall, which can progress to stenosis and food impactions [28]. 

## 2. Eosinophilic Esophagitis (EoE)

### 2.1. Epidemiology, Physiopathology, and Clinical Manifestations

Eosinophilic Esophagitis (EoE) is the best-known EGID, characterized by its exclusive impact on the esophagus [29]. 

EoE was first described in a report by Attwood et al. in 1993, and over the last 30 years, researchers have expressed significant interest in this disease [30].

Advances in knowledge have led to an increase in the recognized prevalence of the disorder, particularly in developed countries. This has resulted in EoE being reclassified from a rare condition to one with significant incidence [27]. The most recent data estimate the cumulative prevalence of EoE in Western countries at 34.4 cases per 100,000 inhabitants, with a slightly higher incidence in the US and Canada (41 cases per 100,000 inhabitants) compared to European countries (29 cases per 100,000 inhabitants) [31]. 

EoE predominantly affects males, with a gender ratio of 3:1 compared to females [32]. It has a hereditary component, with an increased relative risk among siblings, especially monozygotic twins [33]. The etiology of EoE is multifactorial, involving factors such as allergic diathesis, atopy, environmental influences, and genetics [34,35]. Interestingly, a potential inverse relationship with Helicobacter Pylori infection has been reported [36,37]. 

The pathogenesis of EoE is driven by a Th2 inflammation cascade, potentially triggered by food and aero-allergen antigens. This process leads to the production of interleukins, including IL-5 and IL-13, which drive the accumulation of eosinophils in the esophageal wall [38]. The central role of IL-13 in EoE pathogenesis has been highlighted by several trials involving IL-13 monoclonal antibodies [39,40], which have demonstrated correlations with improvements in clinical, endoscopic, and histological outcomes. Additionally, eosinophil chemokines such as Eotaxin-1 and Eotaxin-3 play a central role in eosinophil recruitment and degranulation [41,42]. 

The genetic basis of EoE has been well established, with the identification of the “EoE transcriptome” featuring dysregulated genes, such as CCL26 (encoding Eotaxin-3) [43] and CAPN-14 [44]. Genome-Wide Association Studies (GWASs) have identified other gene risk loci, including EMSY, LRRC32, STAT6, and ANKRD27 [45]. Recent research has also highlighted the role of Tetraspanin 12 (TSPAN12) in advanced fibrostenotic EoE, opening avenues for new therapeutic approaches. Additionally, the role of the microbiota in EoE pathogenesis has been explored [28]. The complex interplay between mucosal-associated microbiota has been studied in relation to several GI diseases, uncovering intriguing correlations [46,47,48,49]. A recent meta-analysis conducted by Massimino et al. has unveiled an underlying dysbiosis, characterized by a shift towards an oral-shaped Gram-negative predominant environment, marking the onset of EoE [50]. 

Clinically, EoE is characterized by a diverse range of symptoms. In adults, dysphagia and food impaction are the most common experiences associated with the condition, while children often present with heartburn, regurgitation, and feeding intolerance [51]. Additionally, the disease can lead to the presentation of a variety of symptoms, including vomiting, nausea, and chest or abdominal pain, which can occur at any age [52,53,54]. EoE can lead to several compensatory eating behaviors, such as avoiding certain foods, picky or slow eating, meticulous chewing, increasing fluid intake during meals, and cutting food into small pieces. These adaptions can reduce the patients’ quality of life [55]. 

Diagnostic delay represents a significant issue with EoE and often leads patients to initially present with food impaction, which is a sign that the disease has already progressed to esophageal remodeling and fibrotic stenoses [52,56]. 

Over the years, numerous symptomatic scores have been developed to standardize the clinical evaluation of EoE [57]. The Dysphagia Symptom Questionnaire (DSQ) and the Eosinophilic Esophagitis symptom Activity Index (EEsAI) are the tools most commonly used to collect information about patient-reported outcomes (PROs), especially in clinical trials [58,59]. More recently, a simplified score, the Dysphagia Days (DD) tool, was introduced by Hirano and colleagues [60]. Accurate assessment of EoE patients, especially when identifying early cases in patients with vague or non-specific clinical presentations, is crucial in reducing diagnostic delays. To aid in this effort, recent advancements in Artificial Intelligence (AI) have led to the development of tools that can detect potential EoE cases using simple clinical and historical data [61].

### 2.2. Endoscopy

Endoscopic evaluation is crucial for the management of EoE, and can be used for diagnosis, disease monitoring, and therapeutic purposes. 

#### 2.2.1. EoE Diagnosis

The Eosinophilic Esophagitis Endoscopic Reference Score (EREFS), developed by Hirano and colleagues, is a validated tool that effectively summarizes the endoscopic traits of EoE (Table 4) [62]. This score includes primary signs like edema, circumferential rings, white exudates, linear furrows, stenosis, and also lists “crepe-paper esophagus” as a secondary feature [62]. 

The implementation of the EREFS has significantly improved the diagnostic capabilities in relation to EoE. Since its introduction, the EREFS has demonstrated high accuracy and inter-operator reliability in distinguishing EoE patients from healthy individuals [63]. Rings, “crepe-paper esophagus”, and white exudates have been identified as the EoE features with the highest agreement between endoscopists [64] (Figure 2). 

However, identifying EoE features can be challenging, as patients in early stages of the disease may lack distinct endoscopic features [30], and approximately 11% of EoE cases present with a normal esophagus [65,66]. A metanalysis including nearly 4600 EoE patients found that while 93% of cases exhibited at least one EoE characteristic feature, the sensitivity for correctly differentiating EoE from other esophageal diagnoses was relatively low, ranging from 15 to 48% [67]. In a retrospective study comparing 151 EoE patients with 226 patients affected by GERD, similar rates of erythema, edema, and erosions, along with normal esophageal appearances, were found in both groups [54].

Current guidelines recommend performing six to eight biopsies in at least two different locations, specifically the proximal and distal esophagus, even in cases in which the esophagus appears normal, when EoE is clinically suspected [68]. Due to the patchy nature of EoE, multiple biopsies throughout the esophagus are necessary to avoid missing eosinophil infiltration. It has been reported that furrows and exudates are the most reliable endoscopic markers of histological inflammation, harboring higher eosinophils concentrations [69,70]. Performing random esophageal biopsies during emergency endoscopies for food impaction, even on healthy mucosa, is strongly recommended [71]. In the case of an EoE diagnosis, this approach allows for early treatment and reduces diagnostic delays. 

Other endoscopic signs, although less common, have been associated with EoE. These include hyperplastic-fibrous inflammatory polyps [72]; the tug-sign, which indicates the need for increased pressure on the forceps when taking a biopsy sample from a site that has previously been biopsied, which is likely related to sub-epithelial fibrosis and remodeling [73]; and the similar biopsy-related pull sign [74]. Also noted is the rare ankylosaurus black sign, which is characterized by a cluster of linear longitudinal whitish nodules often found near a linear furrow [75] and a “caterpillar”-like feature, which is consistently associated with linear furrows [76]. The recognition of these signs can be a critical juncture in diagnostic endoscopy, as they are indicative of potential eosinophilic infiltration, guiding the endoscopist to perform targeted biopsies.

Recent advances in diagnostic endoscopic tools are paving the way for new era in EoE endoscopic assessment [77]. Virtual Chromoendoscopy (VC) techniques have enhanced endoscopic diagnostic performances in multiple settings [78]. In a recent study conducted by Gregory and colleagues, the use of iScan technology (Pentax EC-3490Fi; Pentax, Tokyo, Japan) has shown a sensitivity of 97.6% and specificity of 89.5% in detecting EoE endoscopic features. Interestingly, greater sensitivity and specificity were found for the detection of linear furrows and edema (97.6% and 89.5%, respectively) [79]. The use of Narrow-Band Imaging (NBI) has been linked to an overall enhancement in the diagnostic capability for EoE over traditional white-light endoscopy (WLE). Employing NBI, Tanaka et al. identified three specific endoscopic features typical of active EoE: beige-colored mucosa, dot-shaped intra-epithelial papillary capillary loop (IPCL), and absent sub-mucosal vessels [80]. Remarkably, absent sub-mucosal vascularity had the highest accuracy in differentiating EoE from healthy controls, with a sensitivity of 88% and a specificity of 92%. When combining age, dot IPCLs, and absent submucosal vascularity, the predictions for EoE diagnosis proved to be very reliable (Area Under Curve (AUC) 0.952) [81]. Other VC techniques studied in relation to the diagnosis of EoE include Linked Color Imaging (LCI) and Blue Laser Imaging (BLI). These techniques were used in combination in a study conducted by Abe and colleagues, demonstrating their ability to enhance the visualization of inflamed areas [82]. 

Endocytoscopy, a super-magnifying high-resolution technique, adds another dimension to the range of EoE endoscopic innovations [83]. Early experiences have demonstrated the feasibility of detecting eosinophil infiltration and other histological features such as basal zone thickening, papillary elongation, and spongiosis directly during the endoscopic procedure [84]. 

Over the past decade, AI has brought about a revolutionary change in the field of endoscopy. Machine learning models, a key component of AI, have demonstrated promising results in the detection of EoE-related features. In a study conducted by Rommele and colleagues, an AI model developed and trained on images from 456 patients (61 EoE and 395 controls) showed high predictive values in detecting EoE endoscopic features on a general model of EoE endoscopic features and a more specific EREFS evaluation model, with AUCs of 0.95 and 0.94, respectively, in internal validation. These results were supported by external validation. Remarkably, the AI model surpassed human endoscopists in predictive accuracy, regardless of the endoscopists’ expertise level [85]. 

#### 2.2.2. Therapeutic Drug Monitoring

Endoscopy plays a crucial role in monitoring EoE, since the response to treatments is based on the reduction of the eosinophil count in biopsies [86].

While the EREFS has proven to be invaluable in the diagnostic work-up, limited data regarding its use in therapeutic monitoring have been reported. In a recent prospective study that enrolled EoE patients undergoing various treatment regimens, Dellon et al. found a significant improvement in the EREFS among treatment responders compared to non-responders (mean EREFS from 3.88 to 2.01, *p* > 0.001, with a mean EREFS score of 0.45 for histologic responders versus 3.24 for non-responders, *p* < 0.001) [63]. Additionally, in a post hoc analysis of a Randomized Controlled Trial (RCT) involving patients treated with slurry budesonide, an EREFS score below two was identified as the optimal threshold for therapeutic response [64]. 

The optimal timing for endoscopy in the monitoring algorithm of EoE is not standardized. Current expert recommendation suggest performing a follow-up endoscopy at least six weeks after the latest therapeutic change to assess for histologic changes [87]. However, in clinical practice the timing is often based on the clinical severity of the disease. 

#### 2.2.3. Management of Fibrostenotic Disease 

Uncontrolled EoE causes progressive fibrostenotic remodeling, characterized by smooth muscle hypertrophy and sub-epithelial collagen deposition, as a result of eosinophilic infiltration over time [88]. The endoscopic features of fibrostenotic EoE include esophageal rings and narrowing of the lumen, which can lead to strictures and episodes of food impaction [89]. 

In this context, endoscopy plays a crucial role in providing information to accurately determine the disease stage and guide treatment adjustments. A recent Delphi consensus has established a specific esophageal diameter cut-off of 15 mm to prevent food impaction [87]. However, routine endoscopy may not always precisely assess esophageal narrowing, especially when no distinct rings or stenoses are present [90]. 

Endoscopic Functional Lumen Imaging Probe (EndoFLIP) is a novel technique that has emerged as a valuable tool to assess the caliber of the esophagus and the fibrostenotic evolution of EoE. Lower values of the Distensibility Index (DI) measured by EndoFLIP have been shown to be associated with the fibrostenotic phenotype of EoE, as opposed to inflammatory disease. Interestingly, a lower DI has also been effective in predicting future episodes of food impaction [91]. Recently, Carlson and colleagues developed an activity score for FLIP measurements that correlates with the mucosal eosinophil count and EREFS score [92].

In cases of stricture, endoscopic dilatation (ED) is the standard of care. ED can be performed using either pneumatic balloons or Savary bougies (Figure 3). Regardless of the dilatation method, it is advisable to start with smaller diameters and gradually increase the size of the dilators until a significant mucosal tear or damage is achieved [93]. Given the weakened and fragile state of the esophageal mucosa in EoE patients, it is preferable to schedule ED when histological and inflammatory remission is near completion or has already been achieved [94]. This approach aims to reduce the risk of complications such as perforation and the rare occurrence of Boerhaave’s syndrome [95]. 

The effectiveness of ED in improving symptoms has been highlighted in several studies. Interestingly, no significant differences in efficacy have been reported between balloons or Savary [96]. A meta-analysis that included 27 studies with 845 EoE patients reported symptom improvement in up to 95% of cases (95% CI: 90–98%) after ED. The procedure demonstrated a notably low perforation rate of 0.38% (95% CI: 0.18–0.85%). On average, patients required at least three ED sessions [97]. 

Limited data are available on esophageal stenoses refractory to ED. Biodegradable or Self-Expandable Metal Stents (SEMS) [98,99] appear to be the most suitable option in these cases. However, their placement has been explored in only a few cases within the context of EoE [100]. 

While ED is primarily beneficial for its mechanical effect in widening the esophageal lumen above the 15 mm cut-off, the impact of dilations on the future inflammatory activity of the disease remains an intriguing and not yet fully understood aspect of the disease’s natural history [93]. In the study by Greenberg et al., who enrolled EoE patients with persistent histological activity, implementing a dilation-based approach alongside medical treatment led to better outcomes in terms of symptom control compared to pharmacological treatment [101]. In the future, more data will be needed to fully understand this aspect.

### 2.3. Histology

#### 2.3.1. Histological Features 

Histology is pivotal and is considered the gold standard for EoE diagnosis. The key diagnostic criterion is the presence of an eosinophilic infiltrate exceeding 15 eosinophils per High-Power Field (HPF) or per square millimeter (mm^2^) of tissue. This parameter is often referred to as the Peak Eosinophil Count (PEC) [9]. In clinical practice, eosinophils are counted in esophageal tissue samples, which are formalin-fixed and stained with hematoxylin–-eosin [102].

Counting eosinophils per HPF has traditionally been the standard method of diagnosing EoE. However, this approach presents challenges due to intra-operator and technical differences, including variability in the sizes of microscopic fields across different microscopes. These variabilities can lead to inconsistencies in eosinophil counts, affecting the accuracy and reliability of the diagnosis. To address these challenges, the method of counting eosinophils per mm^2^ was introduced as an alternative. This technique aims to provide a more standardized and precise approach, as the area of a mm^2^ is consistent regardless of the microscope used, mitigating the issue of variability seen for HPF counts. Nevertheless, counting per HPF remains widely used and is a critical component of the diagnostic process.

In recent years, advancements in digital pathology have presented opportunities for further improvements. This process could be further optimized with virtual programs capable of scanning and digitizing esophageal glass slides to facilitate more precise counts [103]. In this scenario, AI tools offer the potential for more accurate and reproducible eosinophil counts, thereby enhancing the reliability of EoE diagnosis. However, in a recent study conducted by Archila et al., a new AI deep-learning tool demonstrated diagnostic accuracy comparable to that of two pathologists [104]. Further developments are anticipated regarding the use of AI in the histological characterization of EoE.

Aside from the eosinophils count, additional characteristic histologic features of esophageal tissue are present in EoE. These include, among others, basal zone hyperplasia (BZH), lamina propria fibrosis (LPF), Surface Epithelial Alterations (SEA), and epithelial dilated intercellular spaces (DIS) [105,106] (Figure 4). While these histological findings are not exclusive to EoE and may be observed in other esophageal conditions, they typically manifest more severely in EoE patients. The presence of these features increases the certainty of an EoE diagnosis [107]. 

Other cell types, primarily lymphocytes and mast cells, are sometimes increased in the esophageal tissues of EoE patients [108].

Eosinophilic infiltrates and other histologic characteristics observed as part of EoE can also be found in various other esophageal conditions, although they are often less pronounced. Conditions such as GERD [109,110], infectious esophagitis, and drug-induced esophagitis can present with similar histological features, including the presence of eosinophils. Additionally, disorders like achalasia and connective-tissue diseases, including scleroderma, may mimic EoE in their histological presentation [27]. Hypereosinophilic Syndrome and Crohn’s Disease can also involve the esophagus [27]. 

From a pathologist’s perspective, a comprehensive histopathologic analysis of esophageal samples, from the surface epithelial layers to the mucosal and submucosal layer, is crucial to guide clinical suspicion. The orientation of the biopsy sample can increase the diagnostic efficacy [111].

Given the patchy nature of EoE, while PEC is often sufficient for diagnosis, there might be cases in which the characteristics of eosinophilic inflammation might not correlate with eosinophil counts that fall below the standardized cutoff. Hiremath and colleagues have reported that DIS is the only histological feature that is evenly distributed through esophageal biopsies from different esophageal locations [112]. 

In cases of suspected EoE in which PEC is not reached, it is important to consider more than just histological data. Therefore, current guidelines recommend that the diagnosis of EoE should integrate histological findings with the clinical presentation of the patient. This approach includes evaluating the patient’s symptoms, clinical history, and additional diagnostic tests, including an endoscopy [9]. Combining histological activity with the clinical profile provides a more comprehensive understanding of the disease. This integrated method helps reduce the risk of misdiagnosis or missed diagnosis.

#### 2.3.2. Scores Assessing Histology 

Scoring systems offer a standardized method to evaluate and quantify the histological features in esophageal biopsies, which is crucial for diagnosing and monitoring the progression or remission of EoE.

The Eosinophilic Esophagitis Histologic Scoring System (EoEHSS), introduced in 2017 by Collins et al., is a prime example of such a system. It includes the histologic characteristics typical of EoE, including eight major histological features, with only two of them being directly eosinophil-linked. Each characteristic is assigned a weighted score, leading to a comprehensive grading system with quantitative descriptions. The EoEHSS also features a staging system based on the percentage of the biopsy specimen affected by these specific alterations [113] (Table 5). EoEHSS has been internally validated, revealing significant differences in treated and untreated EoE patients [113]. Lin and colleagues found higher grading scores in active EoE compared to inactive EoE and GERD, with a notable distinction in the mid and proximal esophagus, but not in the distal portion [114]. 

Several studies have assessed the accuracy of EoE detection, yielding high AUC values of 0.93 in adult EoE patients [37] and 0.92 in children [115]. When comparing EoEHSS with PEC, Ma et al. found similar outcomes in correlating histological EoE activity (AUC of 0.73 in both methods) [116]. In pediatric studies, EoEHSS has been found to be superior to PEC in predicting endoscopic remission [117]. Alexander and colleagues suggested that an EoEHSS score ≤3 correlates with endoscopic and histologic disease remission, but not with symptoms [118]. These findings underscore the importance of a multifaceted approach in managing EoE, where different scoring systems like EoEHSS can be effectively employed to assess disease activity and monitor treatment response. Recently, Collins et al. developed the EoE Histologic Remission Score (EoEHRS), which has highlighted the correlation between histology and the clinical status of patients. Intriguingly, this score also correlates with non-invasive biomarker levels, such as tryptase mRNA and mast cell marker CPA3. These findings have the potential to open up new horizons in the future that will potentially provide a more holistic view of the disease state [119]. 

### 2.4. Treatment: Target Drugs and Emerging Therapies

Proton Pump Inhibitors (PPIs) remain the first line of EoE therapy, even though their use is off-label and clinical and histologic remission rates are limited to 60.8% and 50.5%, respectively, as highlighted by a recent meta-analysis [120]. 

The introduction of an EoE-specific topical steroid, the orally dispersible budesonide tablet (BOT), has revolutionized EoE treatment. BOT has been shown to achieve high histological remission rates (up to 90.1%) and significant clinical response rates (up to 75.1%), with low rates of adverse events [121,122,123]. 

Recently, a novel formulation of orally dispersible Fluticasone has demonstrated optimal histologic, endoscopic, and clinical remission rates comparable to BOT, with these effects persisting through 52 weeks of follow-up [124]. 

In terms of dietary approaches, the most recent meta-analysis by Arias and colleagues reported a histologic success rate of 72% with the six-food-elimination diet (6FED), which was based on excluding milk, wheat, soy, eggs, tree nuts/peanuts, and fish/shellfish, in both children and adults [125]. Other dietary regimens have shown lower efficacy [125]. However, a recent RCT comparing 6FED with 1FED (a milk-based diet) found no significant difference in histologic remission at 6 weeks between the two regimens [126]. 

In recent years, there has been a growing interest in biological therapies for EoE. Dupilumab, a fully human monoclonal antibody targeting IL-4 receptor alpha (IL-4Rα), was the first monoclonal antibody to be approved by the Food and Drug Administration (FDA) and the European Medicines Agency (EMA) for the treatment of EoE. In phase III clinical trials, Dupilumab demonstrated a histologic remission rate up to 60%. Weekly subcutaneous injections have been proven to be the most effective approach [127,128]. 

Other biological targets have shown contrasting results. The IL-13 pathway, targeted by Cendakimab, a humanized anti-IL-13 monoclonal antibody, has reported favorable outcomes compared to placebo in both short and long-term studies [39,40]. Monoclonal antibodies targeting the IL-5 pathway have consistently reduced eosinophil infiltration but have not shown significant improvement in overall symptoms [129,130]. Several other targets are currently under investigation for new therapeutic solutions, including TSLP (NCT05583227), Sphingosine 1-Phosphate [S1P] receptor (NCT04682639), and the KIT pathway associated with mast cells (NCT05774184).

## 3. Eosinophilic Gastritis (EoG) and Enteritis (EoN)

### 3.1. Epidemiology, Physiopathology, and Clinical Aspects

Eosinophilic gastritis (EoG) and eosinophilic enteritis (EoN) are EGIDs characterized by involvement of the gastric or enteric wall, respectively [3,131]. According to TIER 2 nomenclature (for clinical and research purposes), EoN can be subclassified as Eosinophilic duodenitis (EoD), Eosinophilic jejunitis (EoJ) or Eosinophilic ileitis (EoI), depending on the specific segment involved. The term “eosinophilic gastroenteritis” (EoGE) is now used to describe cases involving both the stomach and the small intestine (Table 1) [3]. 

Although much is still unknown about these two disorders, the incidence of both EoG and EoN appears to have increased in recent years [132]. Population-based studies in the United States have estimated the prevalence of EoG at 6.3 per 100,000 individuals and EoG/EoN between 5.1 and 8.4 per 100,000 [133,134]. Interestingly, in contrast to EoE, EoG/EoN tends to affect females slightly more than males [135]. Caucasian ethnicity is associated with a higher prevalence compared to African American and Asian populations [136]. The highest prevalence has been observed in children, with peak incidence noted between the ages of 10 and 24 years, whereas it decreases in older age groups [6,133]. 

The understanding of the physiopathology of EoG and EoN is still developing, with similar mechanisms thought to be involved in both conditions [6,137,138]. Evidence points to a multifactorial etiology involving IgE-mediated allergic mechanisms and delayed cell-mediated responses [139]. Allergic components are suggested by elevated IgE levels and a history of atopy in a significant portion of patients (40–60% of cases) [139,140]. The role of Th2 immunity is becoming clearer [141,142], with genome-wide transcriptome profiling in EoG gastric tissue revealing differential expression of IL-13-driven Th2 immunity pathways, IL-17 signaling, and ErbB- and Wnt-dependent networks [142]. Moreover, gastric biopsy samples have shown significantly increased expression of Th2 cytokines and eotaxin-3 [139,142]. 

No clear correlation between EoG and Helicobacter pylori infection has been found. Hypothetically, the presence of Hp in the digestive tract interferes with the local immune system, triggering an inflammatory response that also promotes eosinophilic gastritis. Eradicating Hp could thus reduce inflammation and provide a more favorable environment for gastric mucosa healing. Some case reports have described the resolution of eosinophilic gastritis with Helicobacter pylori eradication [143,144]. In a Japanese controlled study, which included 22 patients with EoG and EoN, the rate of patients affected by Helicobacter pylori was significantly lower compared to controls (22.7% vs. 48.5%, OR 0.31) [37]. 

EoG and EoN can significantly impact patients’ quality of life [145,146]. 

Clinical manifestations vary and depend on the location of eosinophilic infiltration within the layers of the gastrointestinal wall [147]. Based on these differences, Klein et al. first categorized EoG/EoN disease into mucosal, muscular, and serosal subtypes [148]. 

Mucosal involvement (type 1), defined as eosinophil infiltration of the mucosa and/or mucosal edema, typically manifests as abdominal pain, vomiting, nausea, diarrhea and malabsorption [147,149]. The mucosal subtype tends to follow a chronic progressive course [147], and its symptoms are sometimes misdiagnosed as irritable bowel syndrome (IBS) or functional disorders [145,147]. Studies have linked higher abdominal symptom severity to greater eosinophil levels, especially in the gastric antrum [150]. Imaging can show thickening or nodularity in the antrum and thickened and abnormal enhancement of the small bowel, although these are not specific findings. Two radiologic signs have been associated with bowel involvement: the “aracneid-limb-like” sign, a spider-leg appearance of contrast within the mucosal sinuses resulting from mucosal thickening, and the “halo sign,” which is represented by the layering of the bowel wall due to submucosal edema [151,152,153]. These signs are characteristic of inflammatory pathology and can help differentiate EGIDs from neoplastic conditions such as lymphoma or carcinoma [154,155]. 

Muscular involvement (type 2) can result in thickening and rigidity of the stomach/small bowel, frequently causing intermittent or relapsing obstructive symptoms [147]. Imaging in this subtype may show bowel strictures and decreased luminal diameter, mostly in the distal antrum or proximal small bowel [153]. 

The serosal subtype (type 3), which is the most severe, can lead to exudative ascites [147] and, in some cases, to pleural effusion, mesenteric lymphadenopathy with central necrosis, and other signs of severe inflammation. These symptoms are often more difficult to manage compared to those associated with the mucosal or muscular subtypes. The response to treatment can vary from person to person. While some patients respond well to therapy and may achieve disease remission, others may experience a more chronic or recurrent course [153].

### 3.2. Endoscopy

#### 3.2.1. Eosinophilic Gastritis 

The endoscopic appearance of EoG can vary significantly. Similar to EoE, it is not uncommon for the mucosa to appear normal (Table 6). In the study by Pesek and colleagues, which involved 142 patients affected by EoG, regular gastric mucosa was observed in 62% of cases [132].

Although the literature on this topic is limited, other studies have reported endoscopic abnormalities in the majority of patients. In Hirano’s study, which involved 98 EoG patients, abnormalities of the gastric mucosa were described in 92% of the cases. These included erythema, granularity, erosions (Figure 5), and pyloric stricture [156]. In the study conducted by Lwin et al., erythema and erosions were described as the most frequent endoscopic findings in patients affected by EoG [10]. 

Hirano et al. developed the EoG Endoscopic Reference System (EG-REFS) to standardize the endoscopic assessment of EoG [156]. In the validation study, the EG-REFS score was found to correlate with physicians’ assessments of endoscopy severity. Additionally, a significant correlation was observed between higher EG-REFS severity scores and active eosinophilic gastritis on histology, defined as ≥30 eosinophils in at least five HPFs. However, active histology was found to be associated with regular endoscopic findings in 8% of cases. 

#### 3.2.2. Eosinophilic Enteritis

The ileum is the most frequently affected segment in EoN, with about 30% of EoN cases showing isolated ileum involvement. In a study conducted by Sasaki et al. involving 23 patients affected by EoGE (old nomenclature), ileal erythema was the most common endoscopic finding, which was present in 45.4% of cases and typically exhibited a patchy distribution. Other notable findings included villous atrophy, edema, erosion, and ulcerations, which were occasionally linked with stenosis. Rarer endoscopic appearances included mucosal congestion, whitish exudate, short rounded edematous villi, and dark blue discoloration of the deeper ileal layers in serosa-type cases. Exclusive jejunal involvement was noted in 20% of the cases [157]. 

Other studies have reported a variable range of involvement across the small bowel. In a case series of three patients, lesions were found throughout the entire small bowel [159].

In the study by Pesek et al., which included 123 patients affected by EoGE, the main duodenal and jejunal abnormalities included friability, erythema, nodularity, stricture, and ulcerations [160]. 

It should be noted that in the case of EoN, many patients may present with a normal endoscopic appearance (Table 6). 

#### 3.2.3. Biopsy Sampling 

Given that endoscopic findings are not specific for EoG/EoN, and some patients may present with normal endoscopic findings, performing biopsies is essential in cases of clinical suspicion, as the histologic presence of eosinophils is required for diagnosis [161]. The disease’s patchy nature necessitates multiple biopsies from each affected segment [160,162]. In cases where EoG/EoN is suspected, it is advisable to obtain at least eight gastric biopsies (divided between the antrum and the body) and four duodenal biopsies to enhance the diagnostic accuracy [12]. However, variability in clinical practice is still a recognized issue [163]. 

### 3.3. Histology 

Eosinophils typically reside in the lamina propria of the stomach and small bowel [164]. However, physiological intramucosal eosinophil counts can vary widely due to factors like age, region allergen exposure, and infection history [165]. Research in American children has shown that normal gastric eosinophil counts range from 8 to 11 eosinophils per HPF [166], while Swedish adults exhibit an average gastric eosinophil count of 12/HPF [147]. In the US, gastric eosinophil counts below 9/HPF are generally considered normal [165]. 

Quantitative data on eosinophil distribution in the small bowel are more difficult to interpret, with counts of 18.1 ± 17.0 eosinophils for the bulb, while higher density is observed in the distal regions [167].

The historical lack of clear data on the physiological distribution of eosinophils has made standardizing PEC for the diagnosis of EoG and EoN challenging. There are no universally accepted cut-off values for diagnosis.

The most robust evidence has established pathological thresholds of >30 eosinophils/HPF in five HPF areas for a diagnosis of EoG [10] and ≥20/HPF for duodenal, jejunal, and ileal mucosa for a diagnosis of EoN (Table 3) [12]. For the duodenum, a PEC ≥ 30 in at least three HPFs is considered to be diagnostic [12]. Another study by Reed et al. identified a mean eosinophil count of 30 and 20 eos/HPF, respectively, in gastric and duodenal biopsies as indicative of EGIDs (particularly EoGE) diagnosis [158]. A recent pooled analysis of four prospective studies found that thresholds of 20 eosinophils/HPF in five gastric HPFs and 33 eosinophils/HPF in three duodenal HPFs provided high rates of diagnostic sensitivity and specificity. However, a PEC of 33 eosinophils for the stomach and 37 eosinophils for the duodenum yielded comparable diagnostic accuracy (93% sensitivity and 93% specificity) when a single HPF was examined [11]. A study by Collins et al. identified a PEC cut-off for what was traditionally known as EoGE, with a threshold of ≥52 eos/HPF (Table 3) [13]. 

Beyond the eosinophil count, the eosinophilic infiltrate in EoG/EoN exhibits characteristic microscopic features. In EoG, eosinophilic infiltration of the epithelium is commonly observed around the foveola and within the epithelium itself (Figure 6). The most frequent abnormality is eosinophil degranulation, while cryptitis, crypt abscesses, and involvement of the lamina propria/muscularis are less common [160]. Reactive epithelial changes, chronic gastritis, increased lymphocytes/lymphoid aggregates, peptic duodenitis with gastric metaplasia, or heightened neutrophils are other findings that may be observed. However, elevated eosinophil counts without additional histologic characteristics were found to occur more frequently in EoGE stomach biopsies compared to isolated EoG [160]. In EoN, typical features include villous blunting/inflammation, expanded lymphocytes or aggregates, and eosinophil degranulation [160] (Table 6). An accurate diagnostic approach requires that histology be correlated with clinical symptoms and that other potential causes of eosinophilia be ruled out.

### 3.4. Treatment: Current Drugs and Emerging Novelties

Currently, there are no established guidelines for the treatment of EoG and EoN. Systemic corticosteroids are the standard first-line therapy, but their long-term use is limited by side effects [138,165,168]. Alternative treatments include budesonide, elimination diets (with over a 75% response rate for EoG), leukotriene inhibitors, azathioprine, antihistamines, and mast cell stabilizers [169,170,171,172,173,174]. Efficacy trials for Benralizumab, Antolimab, and Lirentelimab (anti-Siglec-8 antibodies) have provided mixed results, showing histologic responses but not consistent symptomatic improvement [12,137,174,175]. A small study showed that Vedolizumab (an anti-α4β7 integrin antibody) improved histology and reduced steroid dependence in some EoG/EoN patients [176]. Cendakimab (an anti-IL-13 antibody) and Dupilumab (an anti-IL-4Rα antibody) are two potential future alternatives, with Phase III and Phase II trials, respectively, currently underway [177,178]. 

## 4. Eosinophilic Colitis (EoC)

### 4.1. Epidemiology, Physiopathology, and Clinical Manifestations

Eosinophilic Colitis (EoC) is an uncommon EGID characterized by eosinophilic infiltration of the colonic wall. According to the most recent definition, the diagnosis of EoC is reserved for patients who present with typical histological features coupled with corresponding symptoms [179]. Conversely, asymptomatic patients with eosinophilic infiltration in the colon are diagnosed with Primary Colonic Eosinophilia (PCE) [180]. 

The incidence of EoC has increased in recent years, primarily due to heightened awareness, yet it remains a rare condition [5]. Recent data have estimated the adult prevalence to be between 2.1 and 3.3 cases per 100,000 individuals [133,134]. The prevalence in the pediatric population is even lower, [133,134,180]. EoC appears to be more prevalent in females and in Caucasian populations, exhibiting a bimodal age distribution in infants and in adulthood, with a mean age at diagnosis of 33.5 years. Gender prevalence data are conflicting: Mansoor et al. reported a higher prevalence in females (2.6 vs. 1.6 per 100,000, *p* < 0.0001) [134]; however, two large population studies conducted over 13 years indicated no gender difference [181,182]. Consistent with other EGIDs, EoC is more common in urban/suburban areas than in rural areas, and among patients with a higher level of education [134,168].

The pathophysiology of EoC remains poorly understood [183]. In a murine model of colitis, granulocyte-macrophage colony-stimulating factor (GM-CSF) has been observed to be responsible for the molecular switch that converts eosinophils from a tissue-protective to a tissue-destructive mode [184,185]. A recent study on colonic biopsies conducted by Shoda et al. revealed that EoC is distinct from other EGIDs, with pathophysiological mechanisms that are not completely dependent on allergic inflammatory reactions [186]. The study identified 987 differentially expressed genes that were overexpressed in EoC tissues, thereby defining the “EoC transcriptome”. 

Interestingly, the pathogenesis of EoC seems to have only a weak correlation with Th2-related allergic pathogenesis. This could characterize EoC as a distinct pathological entity that is different from other EGIDs [186], but further studies are needed on this topic. 

EoC shows a non-specific and heterogeneous clinical presentation. In pediatric patients, it often manifests as chronic, self-limited bloody diarrhea, while adults typically experience chronic abdominal pain and watery diarrhea [25,179,180]. Additional symptoms can include nausea, vomiting, and weight loss. The presence of atopy history in EoC patients often complicates the clinical picture with conditions like asthma, food allergies, rhinitis, or eczema. 

The depth of eosinophilic infiltration in the colonic wall allows for the identification of three disease patterns [147]. 

The mucosal involvement (type 1), defined as eosinophil infiltration of the mucosa, is the most common. This type often follows a continuous disease course (>6 months) without remission, with patients exhibiting symptoms such as bloody diarrhea, microcytic iron-deficiency anemia, and protein-losing enteropathy [5].

Transmural involvement (type 2) occurs when eosinophils infiltrate the muscular layer. It is associated with symptoms like abdominal spasms, pain, and a possible impact on intestinal motility. Complications such as intestinal obstructions, strictures, volvulus, and perforations may occur. The course of the disease in this form is typically recurrent [5]. 

The serosal subtype (type 3) is the rarest and occurs when eosinophilic infiltration reaches the serosa. This form can be associated with more severe symptoms, including eosinophilic ascites and intense abdominal pain [5]. 

Radiological signs include intestinal wall and mucosal fold thickening and submucosal edema [150,153]. Mucosal thickening, stenosis, and sub-mucosal edema form the basis of the “halo sign,” characteristic of EoC. The “arachnoid limb-like sign” may also be observed via radiological imaging [150]. In cases of transmural involvement, stenosis, particularly at the cecum, may be observed.

### 4.2. Endoscopy 

In approximately 70% of EoC cases, no alterations in the colonic mucosa are observed during endoscopic evaluation [183,187,188]. However, when abnormalities are present, they typically involve the colon segmentally, with only about 10% of patients presenting with pancolitis [14,25,183].

The endoscopic findings of EoC are often non-specific (Figure 7) and do not correlate with the severity of symptoms [7]. However, there is a scarcity of available data in the literature (Table 7).

In a study conducted by Turner et al., which included 194 patients with EoC, the most common endoscopic finding was mucosal erythema, followed by erosions, mucosal granularity, and aphthous ulcers (2.6%) [14]. In a retrospective series involving 38 children with EoC, erythema, lymph follicles, and a loss of vascular pattern were found in up to 33% of subjects, and in 11% of cases of pancolitis [189]. Del Arco and colleagues described that among a cohort of 106 patients with a histological diagnosis of colonic eosinophilia, 14.5% of cases showed endoscopically non-specific colitis [187]. In a single-center retrospective series of 37 adult patients, Macaigne G et al., observed that among those showing endoscopic signs, mucosal erythema, mucosal edema with loss of vascular pattern, erosions, and ulcers were the most frequent [183]. Interestingly, none of the patients exhibited involvement of the ileum and rectum. Similar to Ulcerative Colitis (UC), pseudo-polyps can also occasionally be present in EoC [14]. 

Guidelines for non-IBD colitis recommend performing at least two biopsies for each of the five colon segments (the right colon, transverse colon, descending colon, sigmoid, and rectum), and biopsies of any visible abnormalities to rule out EoC in cases of clinical suspicion. Additionally, performing a biopsy of at least two samples from the terminal ileum is strongly advised [188,190].

### 4.3. Histology

The gold standard for diagnosing EoC is histological examination. However, there is uncertainty regarding the precise count cut-off that distinguishes normal from eosinophilic pathology. This arises from the limited number of studies on normal eosinophil counts in the colon [18]. The largest published series indicates that the highest eosinophil counts are typically found in the cecum and in the ascending colon, unlike in the distal colon, where they appear in less than 5% of biopsy samples [21,191]. Kiss et al.’s meta-analysis found that the eosinophil count in the terminal ileum and the colon/rectum for the healthy pediatric population averaged 11.52 and 11.10 eosinophils per HPF, respectively [192].

Diagnosing EoC is challenging because various pathologies can lead to the presentation of colonic eosinophilia and similar symptoms. This highlights the importance of interpreting pathological data in conjunction with clinical conditions.

In the pediatric population, allergic colitis and proctitis are the most common causes of colonic eosinophilia. Rectal biopsies in these cases can show more than 60 eosinophils per 10 HPFs in the lamina propria [193]. Drug-induced colitis, triggered by antiplatelet drugs (clopidogrel, aspirin, and ticlopidine), Non-Steroidal Anti-Inflammatory Drug (NSAIDs) (especially ibuprofen), and estrogenic-progestogen agents is another cause of colonic hypereosinophilia [194,195]. In these cases, left-side colitis is the most common presentation [194]. Colonic eosinophilia is also observed in patients with Inflammatory Bowel Disease (IBD), playing a significant prognostic role. In patients with Crohn’s Disease (CD), a high eosinophil count in the lamina propria correlates with imminent clinical exacerbation [196], while in patients with UC, mucosal and blood hypereosinophilia is linked with increased clinical severity of the disease at diagnosis and a need for an immediate therapeutic step-up [197]. Eosinophilic colonic infiltrates are also seen in biopsies of patients with microscopic colitis, including both lymphocytic and collagenous forms [198,199]. Among connective tissue diseases, rheumatoid arthritis uniquely exhibits patterns of colonic eosinophilia [200]. In vasculitis patients, such as those with polyarteritis and eosinophilic granulomatosis with polyangiitis, eosinophilic inflammation is evident in the small colonic vessels [201]. 

In EoC, the eosinophilic inflammatory infiltrate can be present in various layers, with the most common location being the lamina propria of the intestinal wall. The depth of the infiltration has clinical implications, as described in the dedicated section. 

According to the strongest evidence, a PEC with more than 50 eosinophils per HPF in the right colon, more than 35 eos/HPF in the transverse colon, or more than 25 eos/HPF in the left colon, along with a consistent clinical and symptomatic profile, indicates a diagnosis of EoC [14] (Table 3). 

In addition to hypereosinophilia, other microscopic characteristics that typify EoC histology include extensive degranulation, eosinophilic micro-abscesses, architectural distortion, fibrosis with mucosal atrophy, loss of mucin, and follicular lymphoid hyperplasia, often accompanied by lymphocytes and plasma cells (Table 7) [187,188].

### 4.4. Treatment

Different approaches are required for the management of EoC in pediatric and adult patients. For children, dietary avoidance is often the primary method of treatment. This typically involves the use of amino acid-based elemental formula or semi-elemental diets. Allergy test-based food exclusion diets are also employed to a lesser extent. In contrast, adults typically receive steroid anti-inflammatory therapy, such as prednisone or budesonide, as the initial treatment [170,180]. If adults experience a relapse after discontinuing prednisone, indicating steroid-dependent disease, Budesonide Controlled Ileal Release (CIR) can be an effective maintenance therapy. Budesonide CIR has the advantage of primarily topical activity, minimizing the long-term adverse effects associated with steroids [202].

Immunomodulators like azathioprine and methotrexate also represent alternatives to maintenance therapy for EoC. Additionally, Montelukast, a leukotriene receptor antagonist, is beneficial in maintenance therapy due to its ability to block eosinophil homeostasis and prevent their infiltration into the intestinal wall [131]. Fecal microbiota transplantation has been suggested as a rescue strategy in EoC, though this evidence is limited to case reports [203]. Emerging therapies for EoC have mainly been tested in animal models. Studies evaluating the efficacy of anti-Siglec-F antibodies (targeting a sialic acid-binding immunoglobulin superfamily receptor) and anti-CCR3 (cysteine–cysteine chemokine receptor 3) antibodies have shown promising results [204,205]. Future therapeutic options are anticipated with the validation of biological drugs like dupilumab, reslizumab, and mepolizumab, which are currently undergoing testing. 

## 5. Conclusions and Future Directions

EGIDs are a diverse group of pleiotropic and heterogenous diseases that are currently undergoing an increase in epidemiological growth and scientific interest. EoE stands as the most well-known condition in this group, yet the knowledge surrounding EoG/EoN and EoC is rapidly expanding. The rising prevalence of EGIDs in Western countries highlights the necessity for more refined and scientifically robust management strategies. Despite the growing clinical significance of EoG, EoN, and EoC, there is still a lack of standardized diagnostic and therapeutic guidelines for these conditions, underscoring the importance of further exploration into their etiopathogenesis to enhance our understanding and management.

Endoscopy remains a cornerstone for diagnosis, monitoring disease progression, and assessing the treatment efficacy of EGIDs. 

In EoE, the use of the EREFS score has proven accurate for raising diagnostic suspicion and monitoring therapeutic responses. The use of endoscopy with high-resolution visualization tools, particularly virtual chromo-endoscopy, shows promise in improving diagnostic accuracy. Looking forward, the application of emerging techniques may greatly enhance the diagnostic accuracy of endoscopy for EoE patients. Techniques such as confocal laser endomicroscopy and endocytoscopy offer potential for real-time histological insights during procedures, bridging the gap between macroscopic and microscopic examination. Furthermore, advancements in AI and machine learning could provide algorithms capable of identifying various patterns and correlating them with clinical outcomes.

A thorough endoscopic examination is crucial in accurately assessing the esophageal lumen caliber in EoE patients. EndoFLIP technology has already shown potential in providing additional insights into the monitoring of EoE, especially concerning the fibrostenotic evolution of the disease. More research is required to fully establish the role of EndoFLIP in the endoscopic assessment and treatment of EoE. Currently, there is an ongoing RCT (NCT06101095) evaluating EndoFLIP in EoE patients. New data relating to this topic are sorely needed. Additionally, in the context of the endoscopic treatment of esophageal strictures, there is a need for RCTs to explore the optimal management of ED, including the number of sessions required, the timing between dilations, and the type of dilators to use.

For EoG/EoN, characterizing endoscopic findings with additional validated scores is crucial for confirming disease suspicion. Implementing virtual chromoendoscopy techniques and AI in assessments could also be significant for these diseases.

Histology remains a key component in the diagnosis and treatment of EGIDs. Quantifying eosinophils in tissue sections continues to be a cornerstone in diagnosing these disorders. For EoG/EoN, there is an urgent need to clearly define diagnostic thresholds. Investigating the correlation between eosinophil infiltration changes and clinical activity, analyzing further microscopic changes beyond the eosinophil count, and studying other cells involved in the inflammatory response are future research directions in the field of EoE histology. Advances in AI and machine learning promise to revolutionize histological analysis, as systems will become capable of detecting subtle patterns. Furthermore, a deeper understanding of the molecular aspects of EGIDs will facilitate personalized medicine approaches. Traditional histological methods should be complemented by molecular profiling and biomarker studies to develop more targeted therapies based on individual patient profiles.

The synergy between endoscopy and histology is fundamental for the diagnosis and management of EGIDs. In the future, the continued integration and refinement of these two modalities will offer the promise of advancing our knowledge of these diseases, leading to progressive improvement of patient outcomes.

## Figures and Tables

**Figure 1 diagnostics-14-00858-f001:**
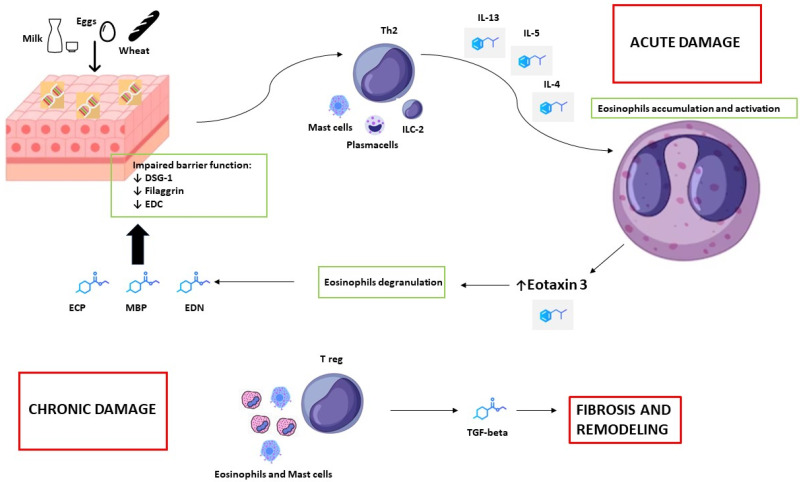
Pathogenesis of Th2 inflammatory drive in Eosinophilic Gastrointestinal Disorders (EGIDs), especially EoE. Exposure to initial food antigens triggers lymphocyte-Th2 activation, resulting in the accumulation of eosinophils in the esophagus. Following stimulation with Eotaxin 3, eosinophil degranulation promotes acute damage to the esophageal epithelium, followed by subsequent chronic fibrotic remodeling of the esophagus, which is dependent on TGF-beta. The copyright of the picture belongs to the authors.

**Figure 2 diagnostics-14-00858-f002:**
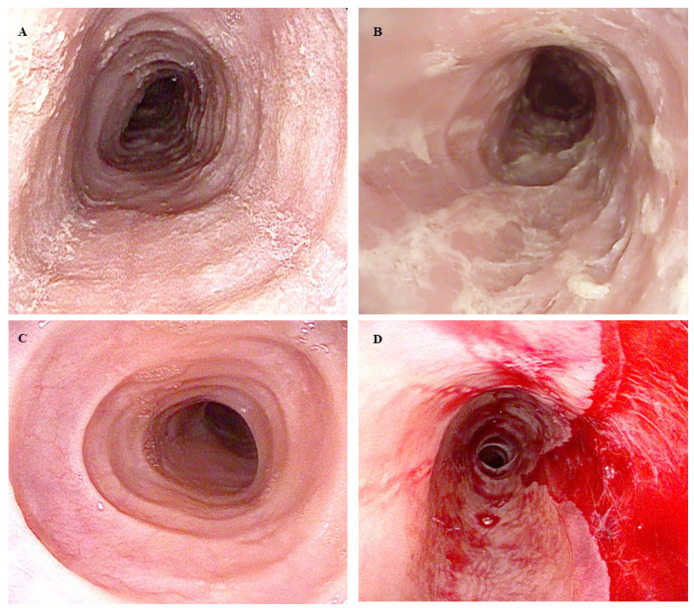
Endoscopic features of Eosinophilic Esophagitis: (**A**) Linear furrows in the middle esophagus. (**B**) White exudates covering more than 10% of the esophageal circumference. (**C**) Prominent rings. (**D**) Noticeable edema, crepe-paper-like appearance, lumen narrowing, and a mucosal tear resulting from endoscope passage. The copyright for the images belongs to the authors.

**Figure 3 diagnostics-14-00858-f003:**
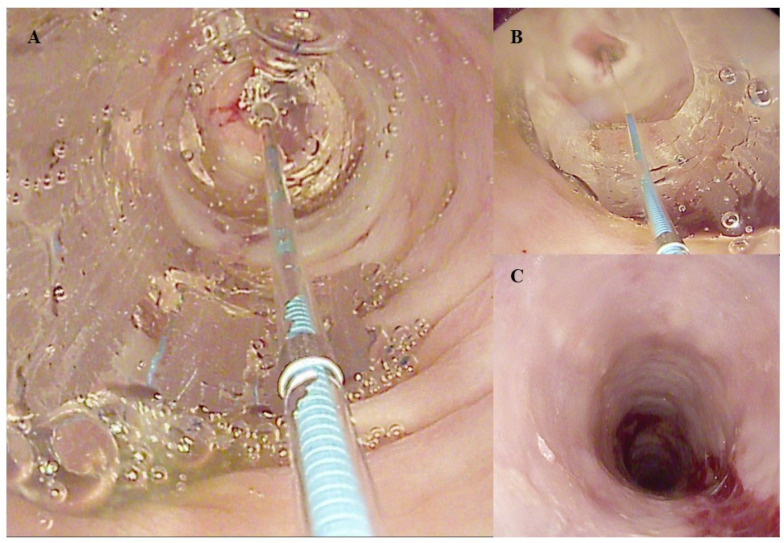
Endoscopic Dilation using a Through-the-Scope (TTS) Balloon for an Eosinophilic Esophagitis (EoE)-related stricture: (**A**) The endoscopic view inside the completely inflated balloon. (**B**) The balloon during deflation. (**C**) The final mucosal tear, indicating efficient dilation. The copyright for the images belongs to the authors.

**Figure 4 diagnostics-14-00858-f004:**
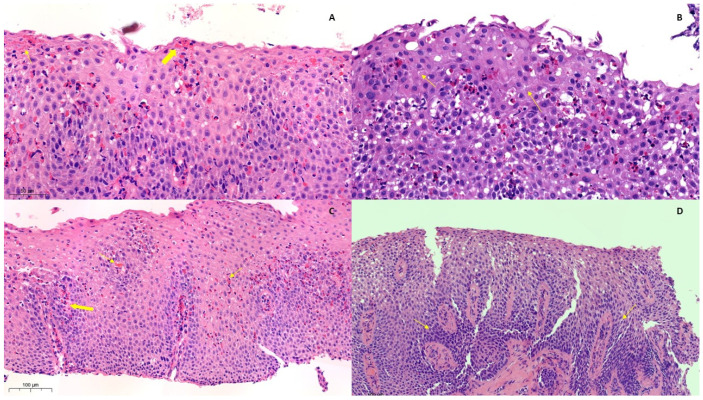
Histological features of Eosinophilic Esophagitis (EoE): (**A**–**C**) Biopsy slides of active eosinophilic esophagitis (EoE) include (**A**) eosinophilic abscesses (thick arrows) and alterations to the surface epithelium (narrow arrows) (20× zoom), (**B**) dilated intercellular spaces (arrows) (20× zoom), and (**C**) basal zone hyperplasia (thick arrow) with eosinophil infiltration (narrow arrows) (15× zoom). (**D**) In cases in which the EoE is in remission, basal zone hyperplasia and papillary elongation (narrow arrows) are evident. Rare eosinophils are present (15× zoom). The copyright for the images belongs to the authors.

**Figure 5 diagnostics-14-00858-f005:**
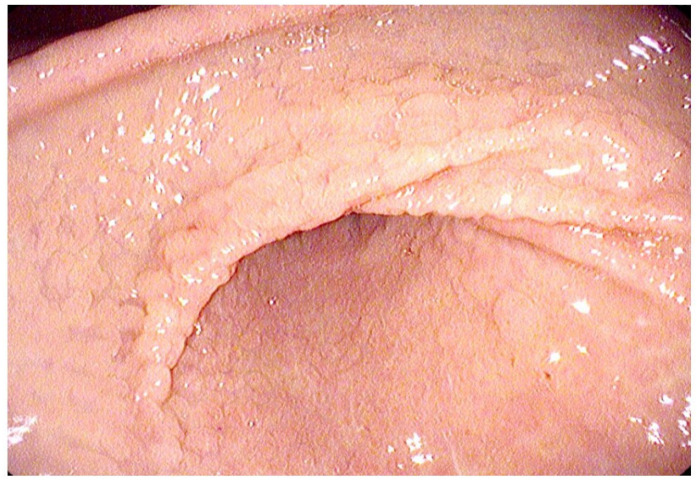
Endoscopic view of active Eosinophilic Gastritis (EoG): fold scalloping with granularity/nodularity in the subangular region.

**Figure 6 diagnostics-14-00858-f006:**
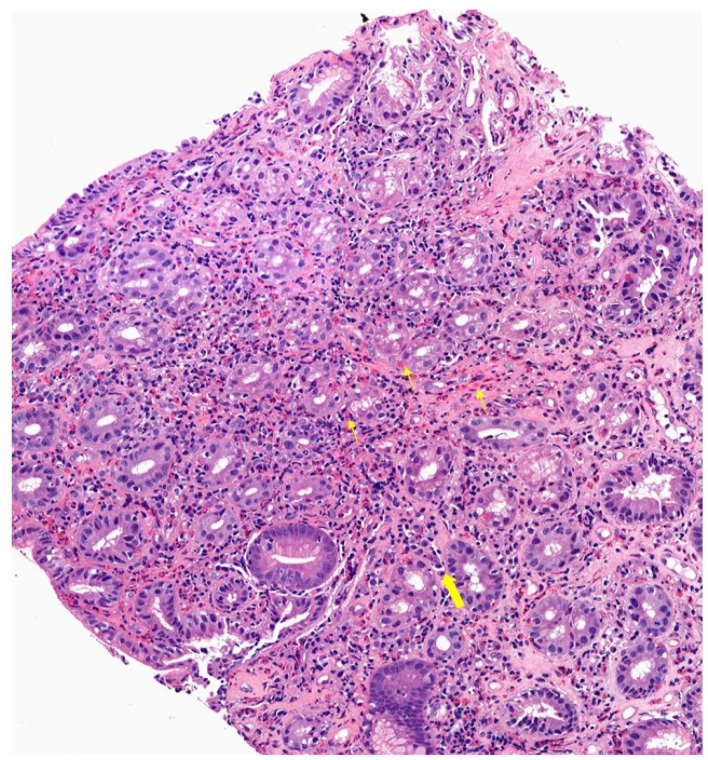
Histological slide of Eosinophilic Gastritis (EoG): Typical findings are eosinophil infiltrates (narrow arrow) and spongiosis (thick arrows) (18× zoom).

**Figure 7 diagnostics-14-00858-f007:**
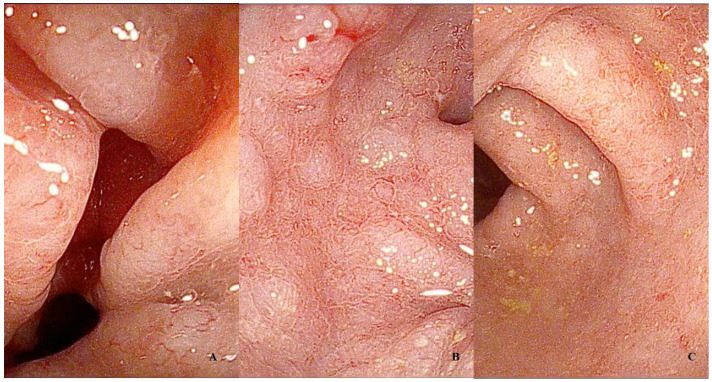
Endoscopic features of Eosinophilic Colitis (EoC): Notable signs are (**A**) erythema, (**B**) loss of vascular pattern, and (**C**) minute erosions.

**Table 1 diagnostics-14-00858-t001:** TIER 1 nomenclature based on international consensus recommendations for eosinophilic gastrointestinal disease nomenclature (2022).

TIER 1 (Clinical Use) Nomenclature	
Esophagus	Eosinophilic Esophagitis (EoE)
Stomach	Eosinophilic Gastritis (EoG)
Small Intestine	Eosinophilic eNteritis (EoN)
Colon	Eosinophilic Colitis (EoC)

**Table 2 diagnostics-14-00858-t002:** TIER 2 nomenclature based on international consensus recommendations for eosinophilic gastrointestinal disease nomenclature (2022).

TIER 2 (Research and Clinical Use) Nomenclature				
	Esophagus	Stomach	Small Intestine *	Colon
Esophagus		Consensus not reached (consider eosinophilic gastritis with esophageal involvement or eosinophilic gastritis and eosinophilic esophagitis)	Eosinophilic gastritis and enteritis Eosinophilic gastritis and duodenitis Eosinophilic gastroenteritis	
Stomach	Consensus not reached (consider eosinophilic gastritis with esophageal involvement or eosinophilic gastritis and eosinophilic esophagitis)			
Small Intestine *		Eosinophilic gastritis and enteritis Eosinophilic gastritis and duodenitis Eosinophilic gastroenteritis		Eosinophilic duodenitis and colitis
Colon			Eosinophilic duodenitis and colitis	

* Can opt to specify known location: Eosinophilic duodenitis (EoD), Eosinophilic jejunitis (EoJ) and Eosinophilic Ileitis (EoI).

**Table 3 diagnostics-14-00858-t003:** Peak eosinophil threshold values for the diagnosis of Eosinophilic Gastrointestinal Disorders (EGIDs).

Disease	Peak Eosinophil Threshold Values	Authors	Year of Publication	Ref.
Eosinophilic Esophagitis	≥15 per HPF (or mm^2^)	Hirano I. et al.	2020	[9]
Eosinophilic Gastritis	≥30 per HPF in ≥5 HPF	Lwin et al.	2011	[10]
	≥37 per HPF for EoD	Dellon et al.	2023	[11]
Eosinophilic eNteritis	≥20/HPF	Dellon E. et al.	2022	[12]
	≥30 per HPF in ≥3 HPF exclusively for EoD			
	≥37/HPF for EoD	Dellon et al.	2023	[11]
	≥52 per HPF for EoGE (old classification)	Collins M. et al.	2009	[13]
Eosinophilic Colitis	Right colon: ≥50 per HPF Transverse colon: ≥35 per HPF Left colon: 25 per HPF	Turner et al.	2017	[14]

HPF: High-Power Field; EoD—Eosinophilic Duodenitis; EoGE—Eosinophilic Gastroenteritis.

**Table 4 diagnostics-14-00858-t004:** Endoscopic EREFS score for eosinophilic esophagitis (EoE).

Endoscopic Feature	Grading	Score
Edema (E)	Absent Mild (reduced vascularity) Severe (absent vascularity)	0 1
Rings (R)	Absent Mild (subtle circumferential ridges) Moderate (distinct rings with easy passage of a standard gastroscope) Severe (distinct rings with impossible passage of a standard gastroscope)	0 1 2 3
Exudates (E)	Absent Mild (<10% of the esophageal area) Moderate/severe (>10% of the esophageal area)	0 1 2
Furrows (F)	Absent Mild (vertical lines with or without depression)	0 1
Stenosis (S)	Absent Present	0 1

**Table 5 diagnostics-14-00858-t005:** Eosinophilic Esophagitis Histologic Scoring System (EoEHSS).

EoEHSS Item	Grading Thresholds	Score	Staging Thresholds	Score
Peak Eosinophil Count (PEC)	Intraepithelial eos not present	0	Intraepithelial eos 0–14 HPF	0
	PEC < 15/HPF	1	PEC ≥ 15 HPF in <33% of HPFs	1
	PEC 15–59 HPF	2	PEC ≥ 15 HPF in 33–66% of HPFs	2
	PEC > 60 HPF	3	PEC ≥ 15 HPF in >66% of HPFs	3
Basal Zone Hyperplasia (BZH)	BZH not present	0	Absent	0
	Basal zone > 15% but <33% of total epithelial thickness	1	BZH (any grade >0) in <33% of Total epithelium	1
	Basal zone 33–66% of total epithelial thickness	2	BZH (any grade >0) in 33–66% of Total epithelium	2
	Basal zone > 66% of total epithelial thickness	3	BZH (any grade >0) in >66% of Total epithelium	3
Eosinophilic Abscesses (EA)	Abscesses not present	0	Absent	0
	4–9 eos aggregates	1	EA (any grade >0) in <33% of total epithelium	1
	10–20 eos aggregates	2	EA (any grade >0) in 33–66% of total epithelium	2
	>20 eos aggregates	3	EA (any grade >0) in >66% of total epithelium	3
Surface Layering (SL)	SL not present	0	Absent	0
	SL of 3–4 eos	1	SL (any grade >0) in <33% of total epithelium	1
	SL 5–10 eos	2	SL (any grade >0) in 33–66% of total epithelium	2
	SL > 10 eos	3	SL (any grade >0) in >66% of total epithelium	3
Dilated Intercellular Spaces (DIS)	IB not present	0	Absent	0
	IB at 400× magnification	1	DIS (any grade >0) in <33% of total epithelium	1
	IB at 200× magnification	2	DIS (any grade >0) in 33–66% of total epithelium	2
	IB at 100× magnification	3	DIS (any grade >0) in >66% of total epithelium	3
Surface Epithelium Alterations (SEA)	SEA not present	0	Absent	0
	SEA with no eos	1	SEA (any grade >0) in <33% of total epithelium	1
	SEA with any eos	2	SEA (any grade >0) in 33–66% of total epithelium	2
	Shed altered surface epithelium admixed with numerous eos consistent with exudate	3	SEA (any grade >0) in >66% of total epithelium	3
Dyskeratotic Epithelial Cells (DEC)	DEC not present	0	Absent	0
	1 DEC/HPF	1	DEC (any grade >0) in <33% of total epithelium	1
	2–5 DEC/HPF	2	DEC (any grade >0) in 33–66% of total epithelium	2
	>5 DEC/HPF	3	DEC (any grade >0) in >66% of total epithelium	3
Lamina Propria Fibrosis (LPF)	LPF not present	0	Absent	0
	Fibers are cohesive; inter-fiber spaces are not demarcated	1	LPF (any grade >0) in <33% of total epithelium	1
	Fibers’ diameter equals basal cells’ nuclei	2	LPF (any grade >0) in 33–66% of total epithelium	2
	Fibers’ diameter exceeds basal cells’ nuclei	3	LPF (any grade >0) in >66% of total epithelium	3

HPF: High-Power Field; PEC: Peak Eosinophil Count; IB: intercellular bridges; Eos: Eosinophils.

**Table 6 diagnostics-14-00858-t006:** Endoscopic and histological features in studies enrolling patients with Eosinophilic Gastritis (EoG), Eosinophilic Enteritis (EoN), or Eosinophilic Gastroenteritis (EoGE).

Author (Year)	Study Design	Population (n)	EGID (n)	Endoscopy	Histology	Ref.
Lwin et al. (2011)	Retrospective	Children (10) Adults (50)	EoG (60)	Regular stomach in 30% of cases. Main abnormalities: erythema (43%), ulcers/erosions (18%), polyps or masses (3%), giant folds (1%), nodular mucosa (1%), gastropathy (1%)	In patients affected by EoG, mean PEC 653 ± 418 eos/mm^2^	[10]
Dellon et al. (2022)	Prospective	Adults (72)	EG (10) EoD (27) EG + EoD (35)	A minimum of 8 biopsies from the stomach and 4 from the duodenum were required to diagnose all 72 cases. Capturing additional cases of EG/EoD incrementally increased with each extra biopsy taken.	EoG: PEC for diagnosis ≥30 eos/HPF in ≥5 HPFs EoD: PEC for diagnosis ≥30 eos/HPF in ≥3 HPFs In patients with EoG, mean PEC 53/HPF In patients with EoD, mean PEC 55/HPF	[12]
Pesek et al. (2019)	Retrospective	Children (317) Adults (including EoC) (56)	EoG (142) EoGE (123)	EoG: regular stomach in 62% of cases. Main abnormalities: erythema (24%), ulcerations (8%), nodularity (8%), mucosal friability (6%), EoGE: Regular stomach, duodenum and jejunem in 66%, 83% and 67% of cases, respectively. Main endoscopic findings: ulcerations (6%), nodularity (3%), erythema (2%), mucosal friability (2%)	EoG: PEC for diagnosis = 87 eos/HPF EoGE: PEC on gastric biopsy for diagnosis = 78 eos/HPF High eosinophils count associated with duodenal abnormalities.	[132]
Hirano et al. (2022)	Prospective	Children (58) Adults (40)	EoG (98)	Erythema (72%), raised lesions (49%), erosions (46%), granularity (35%), thickened folds (26%), mucosal friability (26%), pyloric stenosis 1.5%	Active histology associated with a higher EG-REFS score Active histology (≥30 eos/HPF) associated with regular endoscopic findings in 8% of cases.	[156]
Sasaki et al. (2022)	Systematic Review (16 studies)	Child (1) Adults (23)	EoGE (23)	Isolated ileum involvement in 30% of cases. Findings: redness/erythema (45%), villous atrophy (41%), edema (23%), erosions (27%), ulcerations (27%), stenosis (18%), capsule retention (13%), others (18%).	NA	[157]
Reed et al. (2021)	Retrospective	123 total patients	EGIDs (52) Controls (71)	NA	Controls vs. EGIDs: gastric PEC 3.8 ± 3.6 eos/HPF vs. 5.8 ± 5.0 eos/HPF, duodenal PEC 14.6 ± 8.9 eos/HPF vs. 19.5 ± 11.0 eos/HPF PEC 20 eos/HPF in gastric biopsies or 30 eos/HPF in duodenal biopsies identified EGIDs with 100% specificity.	[158]

EoG: Eosinophilic Gastritis; EoD: Eosinophilic Duodenitis; EoN: Eosinophilic eNteritis; EoEGE: Eosinophilic GastroeNteritis; EGIDs: Eosinophilic Gastrointestinal Disorders; Eos: Eosinophils; HPF: High-Power Field; EG-REFS: Eosinophilic Gastritis Reference Endoscopic Score.

**Table 7 diagnostics-14-00858-t007:** Endoscopic and histological features in studies enrolling patients with Eosinophilic Colitis (EoC).

Author (Year)	Study Design	Population	Mean Age (Years)	Endoscopic Findings	Histology	Ref.
Behjati et al. (2009)	Retrospective	38 EoC children patients	7 (1–14)	Regular colon in 66% of cases. Lymph follicles, erythema, and loss of vascular pattern in 33% of cases. Pancolitis in 11% of cases.	Gradient of eosinophil density decreasing from the caecum to the left colon with relative sparing of the rectum.	[189]
Del Arco et al. (2017)	Retrospective	106 EoC adult patients	50	Regular colon in 68.9% of cases. Other findings: non-specific colitis (14.5%), IBD-like colitis (10.5%)	Mean eos/HPF 43.2 (range 7–199) Findings: intraepithelial eosinophils (67%), eosinophilic abscesses (14.2%), extensive eosinophil degranulation (40.6%), architectural distortion (67%), fibrosis (41.5%), mucosal atrophy (16%), mucosal erosions (5.7%), acute inflammation (16%), lymphoid follicular hyperplasia (23.6%), and lymphoplasmacytic infiltration (26.4%)	[187]
Turner et al. (2017)	Retrospective	194 EoC adult patients 159 controls	53 (18–83)	Regular colon in 32% of cases. Main abnormalities: erythema (11%), erosions (6%), mucosal granularity (6%), aphthous ulcers (3%)	In patients with EoC, mean PEC 166–5050 eos/mm^2^	[14]
Macaigne et al. (2020)	Retrospective	37 EoC adult patients	NA	Regular colon in 69% of cases. Erythema, edema + decreased vascularization, erosions, and ulcerations in 88%, 50%, 63%, and 50% of patients with endoscopic abnormalities. The ileum and rectum are never involved.	NA	[183]

EoC: Eosinophilic Colitis; Eos: Eosinophils; HPF: High-Power Field; PEC: Peak Eosinophil Count; NA: not available.

## Data Availability

No new data were generated or analyzed in support of this research.

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
