# Peer review of "The Dual Lens of Endoscopy and Histology in the Diagnosis and Management of Eosinophilic Gastrointestinal Disorders—A Comprehensive Review"

_diagnostics, 2024, doi:10.3390/diagnostics14080858_

Round 1

Reviewer 1 Report

Comments and Suggestions for Authors

This article reviewed the Eosinophilic gastrointestinal disorders (EGIDs), focusing on endoscopic and histological aspects of this disease.

In general, the article is well-written, although several minor revisions are necessary.

I write down a certain comments and points to be revised.

1)    For Table 2, only PEC threshold of each EGIDs is summarized, but I recommend the authors to summarize all histological features of each EGID.

2)    The authors write the histologic features of EoE in Line 343-346 of P12. I recommend the authors to summarize the diagnostic performances of these features including the sensitivities and specificities.

3)    References should be added for the sentences in Line 357-363 of P13.

4)    The detail of 6FED should be written in Line 421 of P16.

5)    I recommend the authors to add some studies which evaluated the relationship between Helicobacter Pylori infection and the occurrence of EoG.

6)    The authors write the common endoscopic features of EoC in Line 675-677 of P25. I recommend the authors to add the sensitivities of these features.

7)    In the treatment section of EoC, the authors write that dietary avoidance is the primary method of treatment for children. Details should be written for this sentence.

8)    In Abstract, complete words for EREFS should be written.

9)    In Table 1, TIER 1 and TIER2 should be separated.

10) In Table 2, reference 162 is duplicated in Eosinophilic Gastritis and Eosinophilic eNteritis.

11) Exudates(E) section in Table3, <10% should be changed into < 10%.

12) For Table 5, the order of the studies should be changed by the study designs or the published years.

13) For Figure2A, a picture without biopsy forceps is preferable.

14) Figure 4, 5, and 7 are too large and re-sizing is necessary.

15) In Figure 6, size bar should be clearly shown.

16) Line 189 in P9, ‘Figure 3’ should be changed into ‘Figure 2’.

17) Line 249 in P10, ‘454 patients’ should be changed into ‘456 patients’.

18) The word ‘Management’ should be changed into ‘Treatment’ in Line 736 of P27.

19) Underline in Line 510 of P19 should be deleted.

Author Response

Please find attached the file "Response letter"

Reviewer 2 Report

Comments and Suggestions for Authors

Well written article. I note that the term "ipereosinophilia" is mentioned a few times in the article. Are the authors referring to hypereosinophilia? This is the more accepted English term in the medical literature. 

Comments on the Quality of English Language

Well written and easy to understand article. 

Author Response

(The authors gave the same response as above.)
